# Comparison of Long-Term Complications of COVID-19 Illness among a Diverse Sample of Children by MIS-C Status

**DOI:** 10.3390/ijerph192013382

**Published:** 2022-10-17

**Authors:** Sarah E. Messiah, Luyu Xie, M. Sunil Mathew, Sumbul Shaikh, Apurva Veeraswamy, Angela Rabi, Jackson Francis, Alejandra Lozano, Clarissa Ronquillo, Valeria Sanchez, Weiheng He, Sitara M. Weerakoon, Nimisha Srikanth, Madeline Borel, Olivia Kapera, Jeffrey Kahn

**Affiliations:** 1University of Texas Health Science Center at Houston, School of Public Health, Dallas Campus, Dallas, TX 75390, USA; 2Center for Pediatric Population Health, School of Public Health, Dallas, TX 75390, USA; 3Children’s Health System of Texas, Dallas, TX 75235, USA; 4Southern Methodist University, Dallas, TX 75205, USA; 5University of Texas Health Science Center at Houston, School of Public Health, Houston Campus, Houston, TX 77030, USA; 6University of Texas Health Science Center at Houston, School of Public Health, Austin Campus, Austin, TX 78701, USA; 7Texas A&M University, College Station, TX 77843, USA; 8University of Texas Southwestern Medical Center, Dallas, TX 75390, USA

**Keywords:** COVID-19, complications, children, adolescents, MIS-C

## Abstract

Most pediatric COVID-19 cases are asymptomatic; however, a small number of children are diagnosed with multisystem inflammatory syndrome in children (MIS-C), a rare but severe condition that is associated with SARS-CoV-2 infection. Persistent symptoms of COVID-19 illness in children diagnosed with/without MIS-C is largely unknown. A retrospective EHR review of patients with COVID-19 illness from one pediatric healthcare system to assess the presence of acute (<30 days) and chronic (≥30, 60–120, and >120 days) long-term COVID symptoms was conducted. Patients/caregivers completed a follow-up survey from March 2021 to January 2022 to assess the presence of long COVID. Results showed that non-MIS-C children (*n* = 286; 54.49% Hispanic; 19.23% non-Hispanic Black; 5.77% other ethnicity; 79.49% government insurance) were younger (mean age 6.43 years [SD 5.95]) versus MIS-C (*n* = 26) children (mean age 9.08 years, [SD 4.86]) (*p* = 0.032). A share of 11.5% of children with MIS-C and 37.8% without MIS-C reported acute long COVID while 26.9% and 15.3% reported chronic long COVID, respectively. Females were almost twice as likely to report long symptoms versus males and those with private insurance were 66% less likely to report long symptoms versus those with government insurance. In conclusion, a substantial proportion of ethnically diverse children from low resource backgrounds with severe COVID illness are reporting long-term impacts. Findings can inform pediatric professionals about this vulnerable population in post-COVID-19 recovery efforts.

## 1. Introduction

As of 8 October 2022, over 14.8 million children in the United States have tested positive for COVID-19 since the onset of the pandemic [1]. Reports indicate that most COVID-19 cases in children are asymptomatic and do not typically require hospitalization [1]. However, the Centers for Disease Control and Prevention reports that COVID-19 is now one of the top 10 causes of death for children ages 5-to-11 years [2]. Some children with COVID-19 illness are at risk for developing a novel Kawasaki-like disease called multisystem inflammatory syndrome (MIS-C) [2]. As of 1 August 2022, 8798 MIS-C cases have been reported in children ages 5-to-13 years since the pandemic began, most with a positive SARS-CoV-2 test result [3].

Regardless of seriousness of acute infection, some children infected with SARS-CoV-2 can develop symptoms or complications that last for several weeks, but are typically less severe than adults [4]. Symptoms of COVID-19 that persist after acute COVID-19 illness are referred to as long COVID (or post-acute sequelae of SARS-CoV-2 (PASC), post-COVID-19 syndrome, long-haul COVID, chronic COVID syndrome) [4]. To date, long COVID has been shown to consist of over 200 symptoms that cannot be explained alternatively, including fatigue, sleep disturbance, concentration difficulties, loss of appetite, muscle or joint pain, pulmonary fibrosis, myocardial dysfunction, and mental health conditions [4,5,6]. The National Institute for Health and Care Excellent (NICE) guidelines define long COVID as including “both ongoing symptomatic COVID-19 (from 4 to 12 weeks after acute COVID-19) and post-COVID-19 syndrome (12 weeks or more after acute COVID-19) [5]”. At this time, there is no standardized definition of long COVID-19 in adults, and the quality of evidence is still developing [7]. In children, there is a general lack of data regarding long COVID [8], and especially among those diagnosed with MIS-C versus those not diagnosed with MIS-C, and in children less than 11 years old [8]. Yet one recent United Kingdom population-based study suggests that there are potentially tens of thousands of children and young people who might have long COVID [9]. It is important to assess the incidence and prevalence of long COVID in children as it may impact their social, emotional, and mental health and academic performance, as well as their physical health, all of which have the potential to impact a child’s quality of life even beyond the pediatric years [10]. Yet, there are very few studies that have assessed the social and mental impacts of COVID-19 illness in children [11,12]. This is especially true for those children most severely impacted by COVID-19 illness, and from diverse backgrounds. Therefore, the aims of this study were to assess the prevalence of long COVID (1) physical symptoms and (2) social, mental, and academic impacts among a sample of ethnically and socioeconomically diverse children with COVID-19 illness, and to examine any differences among those diagnosed with MIS-C versus those who were not diagnosed with MIS-C. It was hypothesized that those children diagnosed with MIS-C would report more long COVID symptoms and social impacts versus those children not diagnosed with MIS-C.

## 2. Methods

Study Design. A retrospective electronic health record (EHR) review identified patients with previously diagnosed MIS-C from one pediatric healthcare system that serves predominantly Medicaid-dependent families. Patients or caregivers, if the patient was under 18 years of age, completed a survey from March 2021 to January 2022 to assess the presence of long COVID, which is defined as having COVID-related symptoms or health complications for 4 or more weeks after infection [13]. The UTHealth Committee for the Protection of Human Subjects approved all aspects of the current study.

Study Setting. Children’s Health System of Texas (CHST; serving 68% Medicaid and 60% ethnic minority families) is located in Dallas, Texas. All children included in this analysis were CHST patients due to a COVID-19 or MIS-C diagnosis.

Study Participants. Participants included any child aged 0–19 years and who was diagnosed with COVID-19 or MIS-C at CHST’s two main hospitals or ambulatory clinics between 1 March 2020 and 31 January 2022.

Study Procedures. Caregivers who had a child diagnosed positive for COVID-19 or MIS-C at one of CHST’s facilities were contacted by email or with a telephone call to answer a survey about their child’s illness, if they were still experiencing symptoms, and the duration. Questions were also asked about how their COVID-19 illness impacted physical and mental health and academic, sport, and other activity participation and performance. Study staff attempted to contact caregivers a total of 3 times.

Exposures of Interest. Primary exposure of interest was SARS-CoV-2 infection. Office visit and hospital encounter data of all patients who were admitted to CHST were extracted from the EHR including demographic and MIS-C diagnosis.

Primary Outcomes. The primary outcome measure was presence of long COVID physical symptoms and duration. Questions were also asked about the child’s social and academic health.

EHR Measures. All patient comorbidities were extracted from the International Classification of Diseases, Tenth Revision, Clinical Modification (ICD-10-CM) diagnosis codes, and included asthma, anemia, anxiety, cardiac malformation, depressive disorders, diabetes (type 1 and type 2), epilepsy, hypertension, neurodevelopmental disorders, obesity, substance use disorders, and injury/trauma (ICD-10 codes are listed in Appendix A). Similarly, COVID-19-related common symptoms were also determined using ICD-10 codes, including fever (R50.9), cough (R05), nasal congestion (R09.81), shortness of breath (R06.02), diarrhea (R19.7), headache (R51, R51.9), nausea or vomiting (R11.10, R11.11, R11.2), and loss of taste or smell (R43, R43.2, R43.8, R43.9), chills without fever (R68.83), pain/sore throat (R07.0, J02.9), fatigue (R53.83), and muscle pain (R52). A patient could report more than one symptom.

Survey Measures. Patients or their caregivers were queried on a variety of areas including confirming demographics and diagnoses data extracted for the EHR, the impact of COVID-19 on physical and mental health, academic, and sports performance, vaccine hesitancy, and the family’s financial social determinants of health factors. Physical and mental health was evaluated using select questions from the validated Environmental Influences on Child Health Outcomes (ECHO) COVID-19 Questionnaire. Physical health questions asked about COVID-19 symptoms, their duration (acute [<30 days], and chronic [≥30, 60–120, and >120 days]), and if they continue to persist [14]. Mental health questions probed changes in sleep, social interactions, stress, and other behaviors since infection [15]. As this patient population has a low socioeconomic status, the survey asked if the child had health insurance coverage and which type (private, government, or other). Additionally, the family’s financial wellbeing and access to healthcare was assessed using select questions from the Economic Strain Model [16].

### 2.1. Statistical Analysis

T-test, chi-square (*n* > 5), or Fisher’s exact (*n* < 5) analysis were performed to compare baseline characteristics between MIS-C and non-MISC children, which includes age at diagnosis, sex, race/ethnicities (non-Hispanic white [NHW], NHB, Hispanic and other), education (preschool, elementary school, middle school, and high school and above), insurance status (no insurance, government insurance, and private insurance), prior hospitalization and ICU admission, and length of stay. Prevalence and duration of physical symptoms are reported for MIS-C and non-MISC children, respectively. A comparison of the impact of COVID-19 pandemic on social and behavioral health between MIS-C and non-MISC children was performed using chi-square or Fisher’s exact tests. Crude odds ratios and adjusted odds ratios (aOR) were calculated for chronic long-term symptoms (≥30 days vs. <30 days) to explore significant predictors. Covariates of interests include age, sex, race/ethnicity, insurance status, prior hospitalization (Y/N), prior ICU admission (Y/N), and MIS-C status (Y/N). Two-sided *p*-value <0.05 is considered significant. All statistical analyses were performed using SAS v9.4 (SAS Institute, Cary, NC, USA).

### 2.2. Missing Data and Sensitivity Analysis

No missingness was found for our primary exposure (SARS-CoV-2 infection; *n* = 312). Because the primary outcome (long COVID symptoms) had 15% and 16% missing for MISC and non-MISC children, respectively, we performed a sensitivity analysis to compare patient characteristics by missing status. No statistically significant differences were found for missing vs. non-missing data in terms of age, sex, race, education, hospitalization rate, or length of stay (all *p* > 0.05). Insurance status in non-MISC was the only variable that was influenced by missing data (Appendix A).

### 2.3. Power Analysis

A post hoc power analysis was conducted via the PROC POWER procedure in SAS v9.4. Chronic long-term symptoms (≥30 days vs. <30 days) was entered as the dependent variable following a binomial distribution, and MIS-C status (yes/no) was entered as the covariates following a binomial distribution. The results showed ample power (>0.99) in the logistic regression models.

## 3. Results

A total of 312 patients (52.88% male, mean age at diagnosis 6.65 [SD 5.91] years, 20.51% non-Hispanic white [NHW]) consented to participate and complete survey questions from 15 March 2021 to 7 February 2022. The final analytical sample included 26 MISC children (mean age 9.08 [SD 4.86]) and 286 non-MISC children (mean age 6.43 [SD 5.95]), *p =* 0.032. There was no statistical difference in sex or race/ethnicity between MIS-C and non-MISC groups. More than half of non-MISC children (55.44%) were preschool children and half of MIS-C children were in elementary school. The majority (48.7%) of the sample had government insurance. Nearly all MIS-C patients (96.15%) were hospitalized with a mean length of stay of 7.93 (SD 3.79) days, and 84.27% of non-MISC children were hospitalized with a mean length of stay of 5.12 (SD 5.73) days. Notably, nearly half (42.31%) of MIS-C children were admitted to ICU for an average of 10.13 (SD 3.76) days. (Table 1)

A total of 11 (42.3%) MIS-C patients reported long-term (30 days or more) complications including tiredness (8 [30.8%]), headache (4 [15.4%]), difficulty with thinking/concentration (4 [15.4%]), shortness of breath (3 [11.5%]), anxiety (3 [11.5%]), intermittent fever (2 [7.7%]), chest pain (2 [7.7%]), body aches (2 [7.7%]), hair loss (2 [7.7%]), disrupted sleep (2 [7.7%]), and other symptoms (2 [7.7%]). The prevalence of rash, joint pain, depression, heart palpitation, and loss or change in smell/taste was 3.8%. Non-MIS-C patients reported cough after 30 days. (Figure 1a) About every 1 in 5 non-MISC reported at least one of these previously listed long-term symptoms. The most prevalent symptoms included cough (16 [5.6%]), shortness of breath (15 [5.2%]), tiredness (13 [4.5%]), and disrupted sleep (12 [4.2%]) (Figure 1b).

Figure 2a,b shows the prevalence and duration of physical symptoms among MIS-C and non-MISC children, respectively. A total of 11.5% of children with MIS-C and 37.8% of children without MIS-C reported acute long COVID symptoms and about a quarter of MIS-C and 15.4% non-MISC children reported chronic long COVID symptoms. Specifically, 23.1% MIS-C children felt tiredness after 120 days followed by headache (11.5%), difficulty with thinking (11.5%), and anxiety (11.5%). Interestingly, 30.8% and 25.2% of MIS-C and non-MIS-C children were asymptomatic, respectively.

The majority (93.04%) of our sample reported returning to school or daycare after being ill and more than half were completely in-person. There was no statistical difference in the change in academic or physical activity performance since the pandemic between MIS-C and non-MISC children, and more than half of children reported no change (Appendix A).

Table 2 summarizes caregiver-reported long-term effects on social and behavioral health in MIS-C versus non-MISC children compared to before the COVID-19 pandemic. Non-MISC children were more likely to report increased dietary intake than MIS-C children (34.62% vs. 14.84%, *p* = 0.029). Twenty-eight percent of MIS-C children reported sleeping more while 10.51% of non-MISC children reported sleeping more since the start of the pandemic (*p* = 0.036). Compared to non-MISC children, MIS-C children spent more time with friends either in-person (12.0% vs. 3.72%, *p* = 0.016) or remotely (33.33% vs. 12.17%, *p* = 0.013) since the start of the pandemic. However, more than one-third of MIS-C children reported less social connection. MIS-C children also were more likely to have longer screen time for non-educational purposes than non-MISC children (36.0% vs. 14.58%, *p* = 0.036). Caregivers reported that about 40% of MIS-C children had difficulty in concentrating or angry outbursts since the pandemic, while about 20% non-MISC children reported such changes (*p* < 0.05 for both comparisons).

Results from crude logistic regressions showed age (OR 1.07, 95% CI 1.02–1.12, *p* = 0.004), and being a female (OR 1.78, 95% CI 1.02–3.10, *p* = 0.041) or non-Hispanic Black (OR 2.83, 95% CI 1.16–6.90, *p* = 0.022), were significant predictors for reporting long-term physical symptoms post-COVID illness. After adjustment for age at diagnosis, sex, race, insurance, prior hospitalization, ICU admission, and MIS-C status, the odds of reporting long-term complications increased with age (aOR 1.06, 95% CI 1.01–1.11, *p* = 0.023). Additionally, females were 1.86 times more likely to have long-term complication than boys (aOR 1.86, 95% CI 1.02–3.42, *p* = 0.045). Conversely, children with private insurance were less likely to have long-term symptoms than those with government insurance (aOR 0.34, 95% CI 0.14–0.85, *p* = 0.021). The odds of having long-term symptoms were substantially higher in MIS-C children than non-MISC, but the results were not statistically significant (aOR 1.90, 95% 0.30–2.60, *p* = 0.225) (Figure 3).

## 4. Discussion

This study reports on the prevalence of reported persistent, or long, symptoms after SARS-CoV-2 infection in a sample of ethnically diverse children with government insurance, most of whom were hospitalized with serious illness. Analysis showed that 1 in every 10 (11.5%) children with MIS-C and over one-third (37.8%) of those without MIS-C reported acute (<30 days) symptoms for COVID, while 26.9% and 15.4% reported chronic long COVID, respectively. Similar to adult reports, females were almost twice as likely to report long symptoms versus males, while children from higher resource backgrounds were less likely to report long symptoms. In general, caregivers report that most children returned to their normal daily habits post-COVID illness. However, caregivers reported that about 40% of MIS-C children had difficulty in concentrating or angry outbursts since the pandemic, while about 20% non-MISC children reported such changes. These findings can inform pediatric professionals about this vulnerable population and have important implications for post-COVID-19 recovery efforts, especially as children return to their daily school, activity, and home routines.

There have been limited studies reporting on the prevalence of long-term symptoms of COVID-19 among children, and especially among MIS-C versus non-MISC children who are younger. Our overall prevalence estimates of long COVID in children and adolescents fall in the middle of other studies that report prevalence (1.8–10%) [17,18,19,20] to >30% [9,21]. Those studies that are published indicate that children with prolonged COVID experience fever, cough, muscle pain, headaches, shortness of breath, and sometimes insomnia and heart palpitations [22]. The absence of a standardized definition of pediatric long COVID both in terms of symptoms and time makes this area of inquiry challenging. With studies having various inclusion criteria and follow-up times, there is heterogeneity in research about children with long COVID [23].

Results here showed that females were almost two times more likely to report long COVID versus males, which is similar to adult studies. A recent meta-analysis of long COVID in adults [24] found that among the 18 included studies, 10 included potential risk factors for post-COVID symptoms at one-year follow-up, including sex. Three studies [25,26,27] found that female patients had significantly higher risk of experiencing symptoms one-year post-COVID than males. Other recent systematic reviews have reported similar sex difference findings [28]. The pathophysiologic mechanisms of these sex differences remain unclear and will certainly require more research, as not only did our sample include children, but is also one of the first to include children under the age of 11, so puberty status may be an additional interesting new area of inquiry into long COVID.

One of the unique aspects of our sample is the rich diversity, both in ethnicity and insurance status. While we consider this a strength of the study, it is challenging to compare it to other pediatric studies around the world as many do not report ethnicity or insurance or socioeconomic status of sample participants. Our results showed that non-Hispanic Black children in particular were more than twice as likely to report long COVID versus non-Hispanic white children, but this was not significant in fully adjusted models. Results also showed that having private insurance is protective against long COVID. Similarly, very few adult long COVID studies have analyzed demographic and social determinants of health associated with long COVID. One study in one geographic area in California also found that non-Hispanic Black adults were almost twice as likely to report long COVID versus non-Hispanic whites [29]. Nevertheless, this will remain a critical area of inquiry moving forward, as social determinants of health play an important role in COVID 19 disparities in the short and long term and for both adults and children [30].

Indeed, as the world enters its third year of the pandemic, both abrupt and chronic changes and restrictions have significantly impacted youth. Children have endured lockdowns, remote learning, closure of sports and leisure activities, isolation from friends, hand hygiene, and social distancing. Results here showed a substantial proportion of children diagnosed with MIS-C had difficulty in concentrating or angry outbursts since the pandemic, while about 20% non-MISC children reported such changes. Other studies have shown that changes caused by the pandemic can result in difficulties concentrating, headaches, fatigue, and mood swings [6]. While it is difficult to distinguish the root of symptoms for children, the mental health impact needs to be a priority. The CDC reported a rise in mental health-related ED visits among children between April and October 2020. In just one year, from 2019 to 2020, mental health-related ED visits among children aged 5–11 increased about 24%, and those among 12–17 years increased approximately 31% [10]. A systematic review investigating the relationship between pandemics and children’s mental health concluded that the pediatric population is more likely to face increased rates of depression and anxiety during and after a pandemic. Female adolescents had higher rates of depression and anxiety than male adolescents during the COVID-19 pandemic. In addition, seniors in high school were the most likely to experience depressive and anxiety symptoms, indicating a positive relationship between age and anxiety [11]. Further research in long COVID in the pediatric population can aid in monitoring mental health conditions among children, as well as stress the importance of coping resources in schools to help children deal with challenges, both pandemic and typically adolescents stresses.

### Study Limitations

This study has limitations that need to be mentioned including the lack of a control group and susceptibility to recall or reporting bias, especially in terms of specific symptom status and length. Moreover, our results are based on a limited sample size, especially among children diagnosed with MIS-C which may result in large error estimates for smaller ethnic groups. Finally, selection or participation bias may have resulted in inflated estimates of long COVID. Our limitations are similar to those exemplified in Iqbal’s meta-analysis, with the mean sample size of the studies being about 238 subjects [8]. Another study by Zimmermann [4] found that, of the 14 studies reviewed, only five (36.7%) studies included control groups. Nevertheless, there are very few studies that have assessed the physical, social, and mental impacts of COVID-19 illness in children from ethnically and resource-diverse backgrounds.

Identifying disparities in post-COVID-19 trends among diverse pediatric populations can help guide the allocation of resources and improve health equity in school, healthcare, and community settings [31]. In addition, vaccination efforts should continue to be encouraged among children who are eligible. It will be important to continue to include social determinants of health in future research efforts focused on long COVID in children to target areas that can ultimately reduce health disparities.

## 5. Conclusions

Results here showed that 27% of children with MIS-C and 15% of those without MIS-C reported chronic long COVID symptoms. The three most reported symptoms among children with MIS-C were tiredness, headache, and difficulty thinking, while cough, shortness of breath, and tiredness were the three most commonly reported symptoms in children not diagnosed with MIS-C. Similar to adult reports, females were almost twice as likely to report long symptoms versus males, while children from higher resource backgrounds were less likely to report long symptoms. In general, caregivers report that most children returned to their normal daily habits post-COVID illness. These findings can inform pediatric professionals about this vulnerable population and have important implications for post-COVID-19 recovery efforts, especially as children return to their daily school, activity, and home routines.

## Figures and Tables

**Figure 1 ijerph-19-13382-f001:**
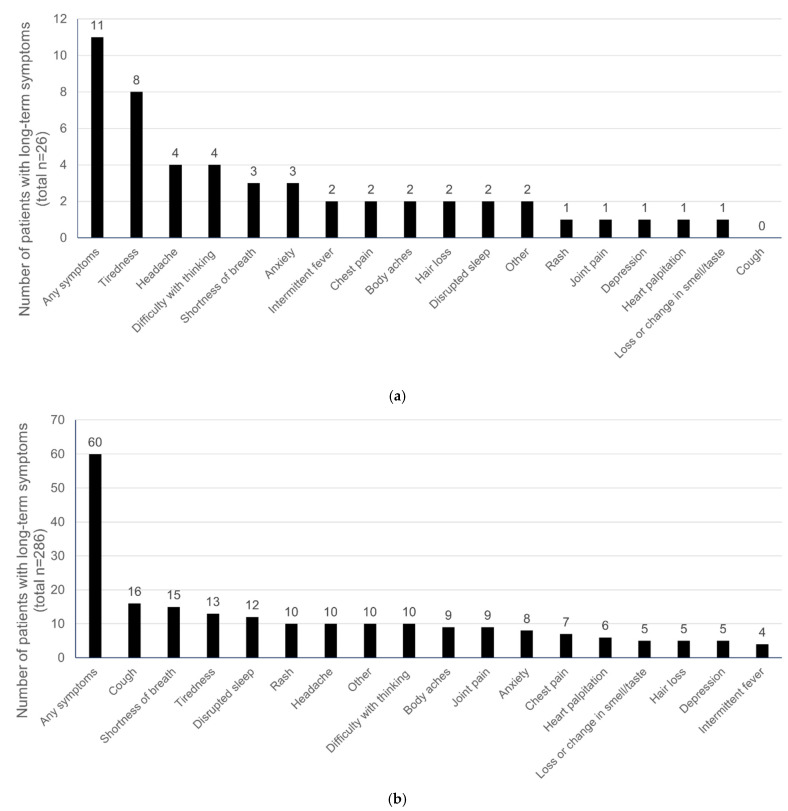
(**a**) Chronic long-term symptoms (≥30 days) in MIS-C children; (**b**) Chronic long-term symptoms (≥30 days) in non-MIS-C children.

**Figure 2 ijerph-19-13382-f002:**
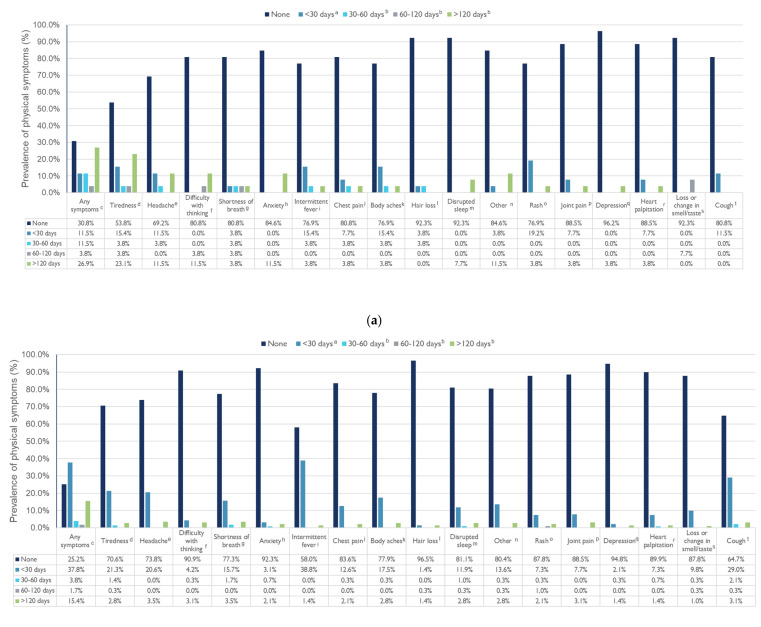
Prevalence and duration of physical symptoms among MIS-C (**a**) and non-MIS-C children (**b**). ^a^ Acute symptoms; ^b^ Chronic long symptoms; ^c^ MIS-C: N_missing_ = 4; non-MIS-C: N_missing_ = 46; ^d^ MIS-C: N_missing_ = 0; non-MIS-C: N_missing_ = 10; ^e^ MIS-C: N_missing_ = 1; non-MIS-C: N_missing_ = 6; ^f^ MIS-C: N_missing_ = 1; non-MIS-C: N_missing_ = 4; ^g^ MIS-C: N_missing_ = 1; non-MIS-C: N_missing_ = 5; ^h^ MIS-C: N_missing_ = 1; non-MIS-C: N_missing_ = 5; ^I^ MIS-C: N_missing_ = 0; non-MIS-C: N_missing_ = 5; ^j^ MIS-C: N_missing_ = 1; non-MIS-C: N_missing_ = 4; ^k^ MIS-C: N_missing_ = 0; non-MIS-C: N_missing_ = 6; ^l^ MIS-C: N_missing_ = 0; non-MIS-C: N_missing_ = 1; ^m^ MIS-C: N_missing_ = 0; non-MIS-C: N_missing_ = 8; ^n^ MIS-C: N_missing_ = 0; non-MIS-C: N_missing_ = 7; ^o^ MIS-C: N_missing_ = 0; non-MIS-C: N_missing_ = 4; ^p^ MIS-C: N_missing_ = 0; non-MIS-C: N_missing_ = 2; ^q^ MIS-C: N_missing_ = 0; non-MIS-C: N_missing_ = 4; ^r^ MIS-C: N_missing_ = 0; non-MIS-C: N_missing_ = 2; ^s^ MIS-C: N_missing_ = 0; non-MIS-C: N_missing_ = 2; ^t^ MIS-C: N_missing_ = 2; non-MIS-C: N_missing_ = 2.

**Figure 3 ijerph-19-13382-f003:**
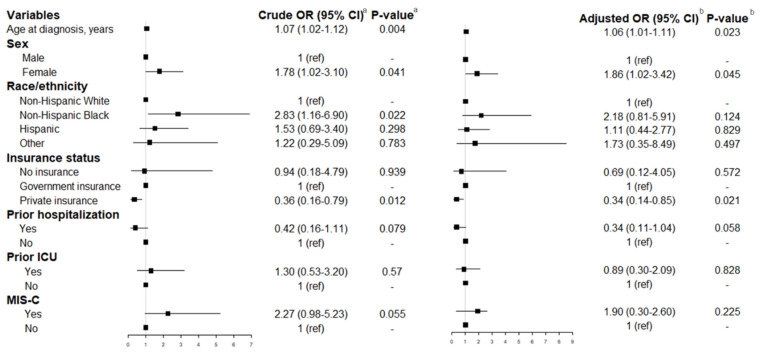
Odds of chronic long-term physical symptoms by demographic and clinical characteristics. ^a^ Crude logistic regression; ^b^ Multivariable logistic regression controlling for age at diagnosis, sex, race, insurance, prior hospitalization, ICU admission, and MIS-C status.

**Table 1 ijerph-19-13382-t001:** Baseline characteristics by MIS-C status (*n* = 312).

	Total (*n* = 312)	MIS-C (*n* = 26)	Non-MISC (*n* = 286)	*p*-Value ^a^
Age at diagnosis, years, mean (SD)	6.65 (5.91)	9.08 (4.86)	6.43 (5.95)	0.032
Boys, *n* (%)	165 (52.88)	13 (50.0)	152 (53.15)	0.839
Race/ethnicity, *n* (%)				0.162
Non-Hispanic White	64 (20.51)	7 (26.92)	57 (19.93)	
Non-Hispanic Black	60 (19.23)	7 (26.92)	53 (18.53)	
Hispanic	170 (54.49)	9 (34.62)	161 (56.29)	
Other/unknown	18 (5.77)	3 (11.54)	15 (5.24)	
Education ^b^				0.001
Preschool	163 (52.41)	5 (19.23)	158 (55.44)	
Elementary school	70 (22.51)	13 (50.0)	57 (20.0)	
Middle school	29 (9.32)	3 (11.54)	26 (9.12)	
High school and above	49 (15.76)	5 (19.23)	44 (15.44)	
Insurance status				0.217
No insurance	8 (2.81)	1 (3.85)	7 (2.70)	
Government insurance	206 (72.28)	15 (57.69)	191 (73.75)	
Private insurance	71 (24.91)	10 (38.46)	61 (23.55)	
Hospitalization, *n* (%) ^b^	266 (85.26)	25 (96.15)	241 (84.27)	0.146
Hospital length of stay, days, mean (SD)	5.77 (5.46)	7.93 (3.79)	5.12 (5.73)	0.083
ICU admission, *n* (%)	28 (8.97)	11 (42.31)	17 (5.94)	<0.001
ICU length of stay, days, mean (SD)	11.0 (6.57)	10.13 (3.76)	11.58 (8.04)	0.639

^a^ T-test for continuous variable and chi-square analysis or Fisher’s exact for categorical variable; ^b^ N_missing_ = 2.

**Table 2 ijerph-19-13382-t002:** Caregiver-reported long-term effect of social and behavioral health in MIS-C versus non-MISC children compared to before COVID-19 pandemic.

Social and Behavioral Health		Total (*n* = 312)	MIS-C (*n* = 26)	Non-MISC (*n* = 286)	*p*-Value ^a^
Dietary intake ^b^	Less	27 (9.57)	3 (11.54)	24 (9.38)	0.029
	Same	208 (73.76)	14 (53.85)	194 (75.78)	
	More	47 (16.67)	38 (14.84)	9 (34.62)	
Sleep patterns ^b^	Less	29 (10.28)	1 (4.0)	28 (10.89)	0.036
	Same	219 (77.66)	17 (68.0)	202 (78.60)	
	More	34 (12.06)	7 (28.0)	27 (10.51)	
Physical activity ^c^	Less	37 (13.45)	7 (26.92)	30 (12.05)	0.058
	Same	216 (78.55)	16 (61.54)	200 (80.32)	
	More	3 (11.54)	3 (11.54)	19 (7.63)	
Spending time outside ^d^	Less	41 (15.30)	6 (24.0)	35 (14.4)	0.326
	Same	200 (74.63)	16 (64.0)	184 (75.72)	
	More	27 (10.07)	3 (12.0)	24 (9.88)	
In-person time with friends ^e^	Less	59 (22.10)	9 (36.0)	50 (20.66)	0.016
	Same	196 (73.41)	13 (52.0)	183 (75.62)	
	More	12 (4.49)	3 (12.0)	9 (3.72)	
Remote time with friends ^f^	Less	22 (8.66)	0 (0)	22 (9.57)	0.013
	Same	196 (77.17)	16 (66.67)	180 (78.26)	
	More	36 (14.17)	8 (33.33)	28 (12.17)	
Screen time for education purposes ^g^	Less	13 (4.98)	0 (0)	13 (5.51)	0.053
	Same	200 (76.63)	16 (64.0)	184 (77.97)	
	More	48 (18.39)	9 (36.0)	39 (16.53)	
Screen time for non-education purposes ^h^	Less	11 (4.15)	0 (0)	11 (4.58)	0.036
	Same	210 (79.25)	16 (64.0)	194 (80.83)	
	More	44 (16.60)	9 (36.0)	35 (14.58)	
Spending time participating in team activities ^i^	Less	26 (13.98)	5 (22.73)	21 (12.80)	0.045
	Same	145 (77.96)	13 (59.09)	132 (80.49)	
	More	15 (8.06)	4 (18.18)	11 (6.71)	
Social connections ^c^	Less	62 (22.55)	9 (34.62)	53 (21.29)	0.045
	No difference	184 (66.91)	12 (46.15)	172 (69.08)	
	More	29 (10.55)	5 (19.23)	24 (9.64)	
Difficulty sleeping ^c^	Not at all	186 (67.64)	15 (62.64)	171 (68.13)	0.673
	Rarely/sometimes	59 (21.45)	7 (29.17)	52 (20.72)	
	Often/very often	30 (10.91)	2 (8.33)	28 (11.16)	
Difficulty concentrating ^j^	Not at all	194 (75.19)	14 (60.87)	180 (76.60)	0.019
	Rarely/sometimes	46 (17.83)	9 (39.13)	37 (15.74)	
	Often/very often	18 (6.98)	0 (0)	18 (7.66)	
Startled easily ^k^	Not at all	211 (77.29)	15 (62.50)	196 (78.71)	0.116
	Rarely/sometimes	49 (17.95)	8 (33.33)	41 (16.47)	
	Often/very often	13 (4.76)	1 (4.17)	12 (4.82)	
Angry outbursts ^c^	Not at all	196 (71.27)	12 (48.0)	184 (73.60)	0.021
	Rarely/sometimes	58 (21.09)	10 (40.0)	48 (19.20)	
	Often/very often	21 (7.64)	3 (12.0)	18 (7.20)	
Slowing down ^l^	Not at all	206 (81.75)	17 (70.83)	189 (82.89)	0.233
	Rarely/sometimes	35 (13.89)	5 (20.83)	30 (13.16)	
	Often/very often	11 (4.37)	2 (8.33)	9 (3.95)	
Felt in a daze ^m^	Not at all	205 (80.39)	18 (75.0)	187 (80.95)	0.233
	Rarely/sometimes	38 (14.90)	6 (25.0)	32 (13.85)	
	Often/very often	12 (4.71)	0 (0)	12 (5.19)	

^a^ Chi-square or Fisher’s exact test; ^b^ N_missing_ = 30; ^c^ N_missing_ = 37; ^d^ N_missing_ = 44; ^e^ N_missing_ = 45; ^f^ N_missing_ = 58; ^g^ N_missing_ = 51; ^h^ N_missing_ = 47; ^i^ N_missing_ = 126; ^j^ N_missing_ = 54; ^k^ N_missing_ = 39; ^l^ N_missing_ = 60; ^m^ N_missing_ = 57.

## Data Availability

Dataset and codes generated during and/or analyzed during the current study are available from the first author on reasonable request.

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
