# Peer review of "Comparison of Long-Term Complications of COVID-19 Illness among a Diverse Sample of Children by MIS-C Status"

_ijerph, 2022, doi:10.3390/ijerph192013382_

Round 1

Reviewer 1 Report

First of all, I would like to congratulate the authors of this article, which is of utmost importance in the field of pediatric health and how pandemics such as COVID 19 are affected by the vulnerable situations experienced by people, both in the field of health care services as well as in mental and social health.

The article meets the theoretical and methodological requirements and will be an empirical reference for health professionals.

I have some suggestions:

1. revise the wording of Figure 1 and in the last line the sentence was left unfinished (row 180).

2. In the conclusions, go deeper into the transcendence of the study and the importance of taking into account these findings in order to differentiate the clinical picture of long covid and multisystem inflammatory syndrome in children.

3. To mention the limitations of the study.

Author Response

Please see attachment that includes a detailed response to both reviewer 1 and reviewer 2.

Thank you

Reviewer 2 Report

In this retrospective study, the authors evaluated patients with COVID-19 illness from one pediatric healthcare system to assess the presence of acute and chronic long-term COVID symptoms

In the present manuscript, they show ethnically diverse children from low-resource backgrounds with severe COVID illness are reporting long-term impacts.

In general, the studies are well done and the manuscript is clearly written.

My comments are shown below, the author may either address these comments or add the limitation in the discussion section.

1.      In the text, the font format and size should be consistent, and should be improved.

2.      Could the authors change some Tables into figures to better show the results?
